# Urine Flow Cytometry Parameter Cannot Safely Predict Contamination of Urine—A Cohort Study of a Swiss Emergency Department Using Machine Learning Techniques

**DOI:** 10.3390/diagnostics12041008

**Published:** 2022-04-16

**Authors:** Martin Müller, Nadine Sägesser, Peter M. Keller, Spyridon Arampatzis, Benedict Steffens, Simone Ehrhard, Alexander B. Leichtle

**Affiliations:** 1Department of Emergency Medicine, Inselspital, Bern University Hospital, University of Bern, 3010 Bern, Switzerland; spiros.arampatzis@hin.ch (S.A.); simone.ehrhard@insel.ch (S.E.); 2University Institute of Clinical Chemistry, Inselspital, Bern University Hospital, University of Bern, 3010 Bern, Switzerland; nadine.saegesser@students.unibe.ch (N.S.); alexander.leichtle@insel.ch (A.B.L.); 3Institute for Infectious Diseases, University of Bern, 3010 Bern, Switzerland; peter.keller@ifik.unibe.ch; 4Institute for Medical Microbiology, Immunology and Hygiene, University of Cologne, 50935 Cologne, Germany; joseph.steffens@uk-koeln.de; 5Center for Artificial Intelligence in Medicine (CAIM), University of Bern, 3010 Bern, Switzerland

**Keywords:** urine analysis, urinary tract infection, UF-4000, automated urine sediment analyser, mixed urine culture, flow cytometry, squamous epithelial cell, prediction, culture growth

## Abstract

Background: Urine flow cytometry (UFC) analyses urine samples and determines parameter counts. We aimed to predict different types of urine culture growth, including mixed growth indicating urine culture contamination. Methods: A retrospective cohort study (07/2017–09/2020) was performed on pairs of urine samples and urine cultures obtained from adult emergency department patients. The dataset was split into a training (75%) and validation set (25%). Statistical analysis was performed using a machine learning approach with extreme gradient boosting to predict urine culture growth types (i.e., negative, positive, and mixed) using UFC parameters obtained by UF-4000, sex, and age. Results: In total, 3835 urine samples were included. Detection of squamous epithelial cells, bacteria, and leukocytes by UFC were associated with the different types of culture growth. We achieved a prediction accuracy of 80% in the three-class approach. Of the *n* = 126 mixed cultures in the validation set, 11.1% were correctly predicted; positive and negative cultures were correctly predicted in 74.0% and 96.3%. Conclusions: Significant bacterial growth can be safely ruled out using UFC parameters. However, positive urine culture growth (rule in) or even mixed culture growth (suggesting contamination) cannot be adequately predicted using UFC parameters alone. Squamous epithelial cells are associated with mixed culture growth.

## 1. Introduction

Urinary tract infection (UTI) is one of the most common reasons leading to an antibiotic prescription for patients seeking ambulatory care in emergency department (ED) visits [1,2,3,4,5]. In particular, in more than 80% of UTI-related consultations, an antibiotic is prescribed [6,7]. As antimicrobial resistance is a continuously growing problem [8], correct diagnosis and the appropriate use of antimicrobial agents in UTI are mandatory.

The gold standard for the diagnosis of a UTI is the detection and identification of the pathogen in relevant quantities by urine culture [9,10,11,12]. However, it is labour intensive with a 24–48-hour time interval between urine sample collection and culture results [13]. Current guidelines recommend the use of empirical antibiotics before culture results are available, which contribute to the rise of multidrug-resistant organisms [14,15].

Examination of urine with urine flow cytometry (UFC) provides good predictions of bacterial growth in cultures [13,16,17]. In addition, the white blood cell count by UFC is an important factor, as pyuria is detected in almost all acute bacterial UTIs [18]. With a throughput of up to 80 urine samples per hour, UFC automated instruments yield results much faster than traditional urine culture methods, thus significantly reducing turnaround time to diagnosis and workload in clinical microbiology laboratories [13,16,19,20,21,22,23].

One challenge in obtaining urine specimens is contamination by urethral, vaginal, and perianal flora [18]. Contamination can lead to polymicrobial growth, a so-called mixed culture. Mixed cultures occur frequently regardless of the collection technique, especially in the busy ED where patient education is often times lacking, and which could not be predicted by urinalysis thus far [24]. Indwelling urinary catheters are associated with a higher rate of mixed culture, especially since 100% are colonised after 14 days [25]. The result of a mixed culture is difficult to interpret because it may be an irrelevant contamination, but it may also be an existing relevant pathogen, making it difficult to adjust the antimicrobial treatment [26].

If the growth of a mixed culture could be predicted by UFC, it could (i) abbreviate diagnosis and therapy, (ii) lead to rapid collection of a new sample with focus on preventing contamination before urine cultures are performed, (iii) save costs associated with growing inconclusive urine cultures, and (iv) reduce the development of resistance because a meaningful antibiogram can optimize antibiotic therapy [27].

New flow cytometers such as the UF-4000 (Sysmex Corporation, Kobe, Japan) are capable of quantifying squamous epithelial cells. Thus far, various studies have shown a correlation between microscopically detected squamous epithelial cells and contamination of urine samples [28,29,30]. However, this correlation is controversial [29,30]. Contaminated urine samples resulting in a mixed culture had a higher count of squamous cells than non-contaminated samples [30]. However, it has also been shown that there is no correlation between squamous cells in urine samples and the growth of mixed flora [13,29]. None of these studies developed a threshold of the amount of squamous cells in urine samples for predicting mixed flora in culture [28,30].

The overall aim of the study is to use UFC parameters to predict different types of urine culture growth. First, we aim to evaluate the association of squamous epithelial cells detected by UFC and the growth of mixed flora in the urine culture using ED real-life data. Second, we aim to develop and validate a statistical model to predict one of the three urine culture growth types: (i) no significant urine bacterial growth, (ii) contamination with the growth of a mixed culture, or (iii) significant urine bacterial growth without evident contamination.

## 2. Materials and Methods

### 2.1. Study Design and Setting

This is a retrospective single-centre cohort study in the ED of the University Hospital of Bern (Inselspital), which is one of the largest EDs in Switzerland, treating about 50,000 patients per year. The study period ranged from 6 July 2017 to 30 September 2020. This study was approved by the ethical committee of the canton of Bern (KEK: 2018-01537) and was performed according to Swiss law. Individual informed consent was waived.

### 2.2. Eligibility Criteria

All patients 16 years of age and older who visited our ED during the study period and whose urine sample was analysed by UFC and urine culture on the same day were included. Spontaneous urine samples including midstream urine (first and last part of the urine stream is discarded, collection of the midstream in sample vessels) and single-use catheter urine were included in the analysis.

Exclusion criteria were (i) analysis of urine sample not with UF-4000 instrument (Sysmex Corporation, Kobe, Japan) with determination of at least bacteria, leukocytes and squamous cells; (ii) lack of urine culture on the same day; (iii) multiple UFC or urine culture analyses on the same day (thus excluding the possibility that UFC analysis and culture facility did not originate from the same urine sample); (iv) paediatric patients under 16 years of age (those are usually treated in a hospital close by); (v) patients who declined further use of their data for research purposes; and vi) urine samples without a clearly defined collection method. Urine from indwelling catheters or replacement bladders, urine after prostate massage, punctured urine and samples explicitly classified as initial stream urine (first part of the urine stream) were also excluded.

### 2.3. Predictors: UFC Parameters

The UF-4000 is a UFC model used for urinalysis in suspected UTIs. The principle is based on the quantitative evaluation of particles or cells according to their size and shape. The individual components pass through a laser beam in a stream of liquid. The light of the laser is scattered in different directions depending on the size and complexity of the particle. It is possible to label specific components of the cells with fluorescently labelled antibodies. In this way, the cells can be distinguished from each other and thus quantified. In this study, the number of (i) leucocytes, (ii) bacteria, and (iii) squamous epithelial cells are used as predictors for the outcome parameters (see below). Additionally, age and sex were evaluated as potential predictors.

### 2.4. Two-Class Classification (Contamination or No Contamination)

First, we investigated the role of squamous epithelial cells in the prediction of mixed culture suggesting contamination of the urine. Thus, the outcome was generated as:

0: no mixed culture—includes all that do not fulfil the definition of a mixed culture; no growth and positive culture.

1: mixed culture (see definitions below).

### 2.5. Three-Class Classification of Urine Culture Growth

Second, we tried to develop a three-class classification of urine culture growth defined as:

0: no growth, negative culture—no significant culture growth (no growth, negative culture) includes everything that is not a mixed or positive culture (see definitions below).

1: positive culture—significant bacterial growth (positive culture) is defined as growth of ≥10^4^ cfu/mL of only one specific germ, as this definition was used by the majority of studies on urine analyses by UFC [31].

2: mixed culture—mixed urine culture growth (mixed culture) is defined in various ways in the literature [16,22,29,32,33]. In this paper, mixed urine culture growth is defined as bacterial growth of ≥10^4^ cfu/mL with a mixed growth pattern (≥2 different germs without predominance of either germ) [16].

### 2.6. Data Collection

Routinely, a urine sample is taken from ED patients suspected of having a UTI. The urine sample is analysed with the UFC, and often, a urine culture is established at the same time. Urine culture results such as species, germ quantity, etc., were provided by the Institute for Infectious Diseases (IFIK, Univ. Bern). General patient data such as age, gender, case number and patient number are stored in the SAP system of the Inselspital (OpenText Suite for SAP1 Solutions, OpenText Corporation, Waterloo, ON, Canada).

#### 2.6.1. Urine Flow Cytometry

All urine flow analyses included in this paper were performed in the ISO 17025 accredited Center for Laboratory Medicine (Inselspital, Bern University Hospital, University of Bern, Bern, Switzerland) using the UF-4000 (Sysmex Suisse, Horgen, Switzerland).

#### 2.6.2. Urine Culture

Urine culture was performed at the clinical microbiology laboratory of IFIK, which is also accredited according to ISO 17025:2018 and which is located on the same campus. The urine samples were transported rapidly in a pneumatic tube system to the microbiology laboratory in sterile, sealed plastic tubes with added stabilising agents (BD Vacutainer; BD, Franklin Lakes, NJ, USA; with boric acid and sodium formiate). In this way, the specimen was stabilised by preventing the multiplication of the existing germs during transport [26]. If the urine sample was transported unrefrigerated for more than 2 hours, it was excluded from the analyses, as a resulting falsification of the pathogen count could not be ruled out [16,18]. After receiving the urine samples in the laboratory, they were processed immediately or stored at 2–8 °C in the refrigerator (when received between 10 p.m. and 7 a.m.). Overall, culture samples were processed within 10 hours after collection. In order to detect any antimicrobial activity (e.g., patients under antibiotic therapy at the time of urine collection), an inhibitor test with Bacillus subtilis was performed. The microbiological cultures were performed on different agar plates [26]. A non-selective medium (Columbia sheep blood agar, in house) and CHROMagar™ Orientation agar (CHROMagar, Paris, France) as selective culture medium were read twice (after incubation for 18 to 24 and 42 to 48 hours at 36 ± 1°C) [18]. Depending on amount and purity, the colonies were identified to species level using MALDI-TOF mass spectrometry and were tested for their antibiotic susceptibility (Kirby Bauer method) according to the European Urinanalysis Guidelines [9] and EUCAST clinical breakpoints (https://www.eucast.org/clinical_breakpoints/ (accessed on 18 February 2022)).

### 2.7. Data Extraction

First, an automatic search was performed to identify patients (i) with UFC using the UF-4000 during the study period, (ii) 16 years of age or older at admission, (iii) consent to anonymous use of their data for research purposes, and (iv) with documented ED admission. Extracted cases were crosschecked and matched with the urine cultures sent to the IFIK. Extraction results were proofed and verified by the Insel Data Science Center (IDSC).

In summary, the following data were extracted:

SAP: patient demographics (age, gender), case ID, patient ID.

IDP (Insel Data Platform): all patient transfer data (consultation ED at the beginning vs. during the stay, date of urine culture and UFC same vs. not same, number of days from admission to urine collection, number of days from urine collection to performance of UFC, number of days and number of hours from admission to performance of UFC, UFC parameters) come from this platform, which concatenates several different systems.

IFIK: germ type, germ quantity, performed analysis, used material, used transport vessel, presence of Legionella antigen or Pneumococcus antigen, notes regarding the analyses (e.g., if the urine has been transported unrefrigerated for more than 2 hours).

The data evaluation was carried out with pseudonymized data. The Insel Data Coordination Lab (IDCL) took over the key management as a trust centre. Persons participating in the study were excluded from key access.

### 2.8. Statistical Analysis

The statistical analysis was performed using R 4.1.1 (R Foundation for Statistical Computing, Vienna, Austria) and Stata® 16.1 (StataCorp, College Station, TX, USA). Categorical variables were described with number (%) and continuous variables with median (interquartile range, IQR). The association of mixed culture with squamous epithelial cells was tested using a logistic regression with the odds ratio and 95% confidence interval (CI) as a measure of strength of the association.

If the machine learning approach showed high predictive ability regarding mixed urine culture with >50% correct classification, a decision tree, a nomogram and cut-off tables for clinical use were developed in analogy to Müller et al., (2018) [16].

#### Machine Learning Approach

A machine learning approach was used to distinguish between the two (mixed culture vs. no mixed culture) and three outcome classes present in the dataset (mixed culture, positive culture, no growth). For this purpose, we used R with the XGBoost (extreme gradient boosting) and caret (classification and regression training) packages as a multi-class capable tool for parameter tuning. The data were divided into a training set (75%) and a test set (25%). Results were visualised using a confusion matrix. In addition to XGBoost, the glm (generalized linear models) and BMA (Bayesian model averaging) packages were used here for the logistic regression. Here, visualisation was performed via ROC curves with the corresponding AUCs. The results are presented with the 95% CI. In addition, a precision recall curve was created. Sensitivity (SE), specificity (SP), positive and negative predictive values (PPV and NPV) in the validation set are calculated for each prediction model.

### 2.9. Ethical Considerations

This study was authorized by the cantonal ethics committee of Bern, Switzerland (KEK: 2018-01537).

## 3. Results

### 3.1. Patient Characteristics

In total, 3835 urine samples were included (Figure 1). The age of the associated patients ranged from 16 to 101 years with an overall median age of 67 (IQR 51-78) years. In total, 45.7% (*n* = 1751) of patients were female and 54.3% (*n* = 2084) were male. In addition, 21.3% (*n* = 817) of the urine cultures fulfilled the criteria for a positive urine culture. A significant mixed culture was found in 13.5% (*n* = 517) of the urine cultures; 65.2% (*n* = 2501) of the cultures showed a non-significant bacterial growth (<10^4^ cfu/mL), i.e., negative urine culture. An overview of the described parameters and other evaluated variables can be found in Table 1. No relevant differences were found between the validation and training sets.

### 3.2. Descriptive Analysis

Figure 2 shows the distribution of the UFC parameter: leucocytes, bacteria, and squamous epithelial cells, according to type of urine culture growth. The distributions in the positive culture and mixed culture group showed a large overlap. Negative cultures had a different distribution of UFC parameter, especially of leucocytes and bacteria.

### 3.3. Prediction of Mixed Culture and the Role of Squamous Epithelial Cells (Two-Class Classification)

Squamous epithelial cells were significantly (*p* < 0.001) associated with mixed culture growth (Table 2). Urine culture growth was significantly associated with sex and age (both *p* < 0.001) with higher rates of positive and mixed cultures in females and geriatric patients (>80 years). Higher rates for females were found in all age groups (Appendix A).

Figure 3 and Figure 4 show three receiver operating curves and precision recall curves, one for each package used: XGBoost, glm, and BMA, to predict mixed urine culture growth out of the predictor variables.

AUC values ranging between 74.4% and 77.9% were achieved. Precision was <40% when recall was >40%.

In total, 10.3% of the mixed cultures and 59.9% of the no mixed cultures were correctly classified (Figure 5). The prediction of a mixed culture had an SE of 78.6%, SP of 68.9%, NPV of 95.5%, and PPV of 27.7% in the validation set.

### 3.4. Three-Class Classification Using Machine Learning

Using XGBoost trained on the training dataset, we achieved a prediction accuracy of 80% in the three-class approach. In a confusion matrix (Figure 6), we visualised the prediction values of (i) mixed cultures, (ii) positive cultures, and (iii) no growth based on leukocytes, bacteria, squamous epithelia, age and sex in the validation set. Table 3 shows the SE, SP, PPV, and NPV of the figures presented in the matrix.

In addition, 1.5% of the datasets was correctly predicted as mixed flora; 6.9% was incorrectly classified as positive cultures and 4.8% as negative cultures, although a mixed flora grew in the urine culture. Thus, of the *n* = 126 mixed cultures, 11.1% (*n* = 14) was correctly predicted (corresponds to the SE), and a total of 88.9% (*n* = 112) was incorrectly predicted as positive cultures (52.4%, *n* = 66) or negative cultures (36.5%, *n* = 46). The PPV was 37.8%.

The prediction of positive cultures showed an SE of 74% and a PPV of 65.3%. Of the *n* = 208 positive cultures, 74% (*n* = 154) was correctly predicted and a total of 26% (*n* = 54) was incorrectly predicted as mixed cultures (7.7%, *n* = 16) or negative cultures (18.3%, *n* = 38).

Of the *n* = 625 cultures without growth, 96.3% was correctly predicted (corresponds to the SE). Furthermore, an SP of 74.9%, a PPV of 87.8%, and an NPV of 91.6% were found (Table 3).

The prediction of a mixed urine culture on the basis of squamous epithelia, leukocytes and bacteria did not have a high predictive ability. Therefore, no new decision-making tools were developed. In 96.6% of all urine samples with UFC parameters in range of the in-house cut-off (leukocytes <25/µL, bacteria <125/µL, squamous epithelial cells <31/µL), no significant urine culture growth was found.

## 4. Discussion

### 4.1. Statement of Principal Findings

In this retrospective cohort study, three types of urine culture growth (no significant bacterial growth, significant bacterial growth without evidence for contamination, mixed urine culture growth) were predicted out of UFC parameters. For this purpose, real-life data of patients consulting the ED were analysed. The cell counts of squamous epithelial cells, leukocytes and bacteria were associated with the different types of growth in urine culture (no significant culture growth, significant bacterial growth, mixed culture growth).

Patient instruction is associated with lower rates of contamination [34]; however, in the busy ED, it is not always possible to properly instruct every patient. Nevertheless, the observed mixed culture type proportion was 13.5%, which is in the range of previously published studies [35]. We achieved a prediction accuracy of 80% in the three-class approach, and predicting “no culture growth” could be performed safely. However, mixed cultures could not be satisfactorily predicted and discriminated from significant bacterial growth. Thus, no new cut-off values or decision tools were developed.

### 4.2. Results in Context

Several studies have tried to establish a link between the presence of different types of epithelial cells and the growth of a mixed flora in urine cultures suggesting contamination [28,29,30,32].

Müller et al. (2018) for instance tried to predict mixed urine flora growth out of epithelial and round epithelial cells analysed by the UFC UX-2000 and failed [16]. The authors concluded that the reason for the prediction failure could be the UFC model used cannot determine squamous epithelial cells, which are thought to have a higher predictive value than the analysed epithelial cells in their study [32]. The hypothesis was supported by the findings of that study, as squamous epithelial cells were significantly associated with mixed culture growth.

Smith et al. (2003) were able to show an association between the presence of squamous epithelial cells in phase contrast microscopy and the number of growing species in urine cultures. Higher numbers in squamous epithelial cells were associated with increased bacterial growth from urine cultures [29].

Walter et al. (1998) detected squamous epithelial cells in 99 out of 105 samples from catheter urine; none of them were bacterially contaminated. In addition, squamous epithelial cells were detected in 101 out of 105 samples from midstream clean-catch urine; only 22 of them were bacterially contaminated. The study showed that squamous epithelial cells are present in both catheter urine and midstream clean-catch urine and are not a good indicator for predicting bacterial contamination [32]. Yang et al. (2017) used the UFC model UF1000i to successfully predict mixed growth (defined as at least two germs) based on bacterial count [33]. Because of these conflicting study results and the new possibilities for analysis with the UF-4000 model, we re-examined the correlations without success. One reason for this difference could be a different patient collective, as Yang et al. (2017) focused on uncomplicated urinary tract infections of women only.

The UFC parameter characteristics including squamous epithelial cells are not enough to differentiate between mixed pattern growth as a sign of contamination and significant bacterial growth.

In our evaluations, the type of urine culture growth was associated with sex and age with higher rates of positive and mixed cultures in females and geriatric patients (>80 years). These findings can be well explained with the results of previous studies. Higher incidences of bacteriuria in females and geriatric patients are found [36,37,38,39]; thus, higher rates of positive cultures have to be expected. Conversely, sex and age group differences regarding higher rates of mixed culture growth in females and geriatric patients might be explained by anatomical differences [38,40] and by variations in specific hormone levels, such as testosterone or estrogen [41,42,43], throughout life and between the sexes. Clinicians should be aware of these findings.

### 4.3. Strengths and Weaknesses of the Study

In this study, microbiological data and laboratory real-life data were analysed. Since the data were analysed retrospectively, urine collection was performed as in normal hospital routine.

The UFC and urine culture data used were obtained by laboratory tests that are regularly validated. Therefore, measurement error of the variables mentioned is unlikely, and high data quality can be assumed. As soon as there was more than one urine collection in one case, the data were excluded because the assignment of the UFC parameters to the correct urine culture could not be performed with absolute certainty. In addition, only datasets were analysed in which UFC was performed on the same day that the urine culture was established. These two selection procedures ensure that the UFC and the urine culture were performed from the same urine sample.

Strengths of the study include the heterogeneity of the ED patients and the large number of patients included. However, not all included patients were treated by the same physician, and there may be a selection bias due to interindividual decisions for or against taking a urine sample, including urine culture.

Another weakness of the study is that no clinical data (e.g., specific UTI diagnosis) were available. The combination of clinical tools and UFC parameters to predict urine culture growth is an important research area for future studies, as cut-off values vary between different patient subgroups [44]. Inclusion of dipstick test results also needs to be investigated in further studies. The presence of leukocyte esterase or nitrite is interpreted with a sensitivity of 75% and a specificity of 82% as a positive result regarding the presence of UTI [45,46]. In combination with UFC parameters and clinical information, further decision-making tools may be developed.

Furthermore, an adjustment of the definition of positive urine culture may be considered in future studies predicting culture growth. Studies comparing urine and bladder aspirate samples in women with cystitis have shown that the traditional criterion for a positive urine culture (10^5^ CFU/ml) is insensitive and that 30–50% of women with cystitis only have 10^2^–10^4^ CFU/ml in voided urine [47,48,49]. Since most clinical laboratories do not quantify bacteria below a threshold of 10^4^ CFU/ml [49], a culture report of no growth in a woman with urinary symptoms should be interpreted with caution.

Last, a mixed culture can in some circumstances be a coinfection of different germs, thus not a contamination, a status which can only be interpreted in the clinical context of a patient.

### 4.4. Implications for Clinicians

Squamous epithelia, bacteria and leukocytes are clearly associated with future culture growth. Prediction of negative cultures and thus exclusion of positive or mixed cultures can be performed with good accuracy [16]. Furthermore, the presence of squamous epithelial cells increases the likelihood of mixed culture growth. Taking into account these findings in clinical practise has the potential to minimize unnecessary antibiotic therapy. Furthermore, Alenkaer et al. (2021) proposed that up to 36% of the urine cultures ordered could be avoided by using UFC [17], which would reduce the workload in the microbiology laboratories and thus costs.

The presented in-house cut-off for bacteria, leucocytes and squamous epithelial cells is a safe way to rule out future culture growth.

Thus, we recommend (i) recollecting of urine samples after patient instruction in the presence of squamous epithelial cells before cultivation to avoid contaminated urine cultures and (ii) not performing a urine culture if the cut-off values of the UFC parameters are within the normal range of predicting no bacterial growth to avoid unnecessary costs.

## 5. Conclusions

Squamous epithelial cells, bacteria and leukocytes are associated with the different types of urine culture growth, especially mixed culture growth. The likelihood of a mixed culture growth increases when the count of squamous epithelial cells is higher. UFC parameters can safely predict a urine culture result without significant bacterial growth (rule out). However, positive urine culture growth (rule in) or even mixed culture growth suggesting contamination cannot be adequately predicted nor differentiated using UFC parameters safely. Clinicians should be aware of these findings to guide therapy and further diagnostical tests.

In order to be able to predict mixed cultures in the future, further analyses are necessary that include additional clinical parameters.

## Figures and Tables

**Figure 1 diagnostics-12-01008-f001:**
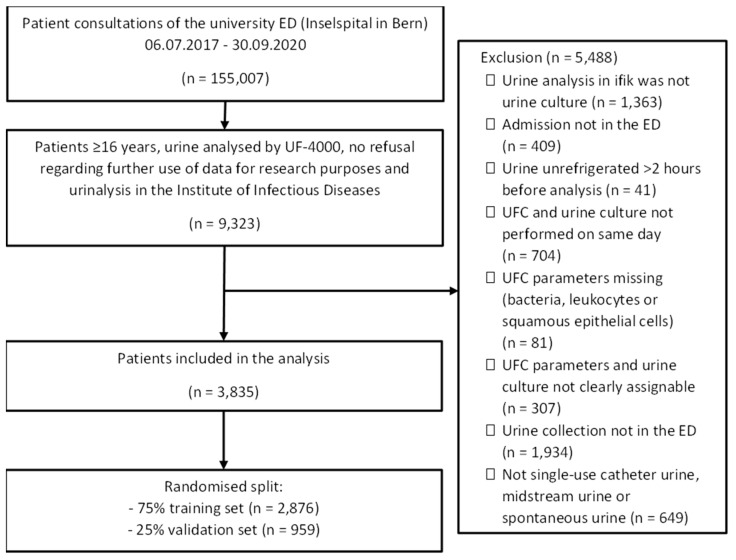
Flowchart of the study.

**Figure 2 diagnostics-12-01008-f002:**
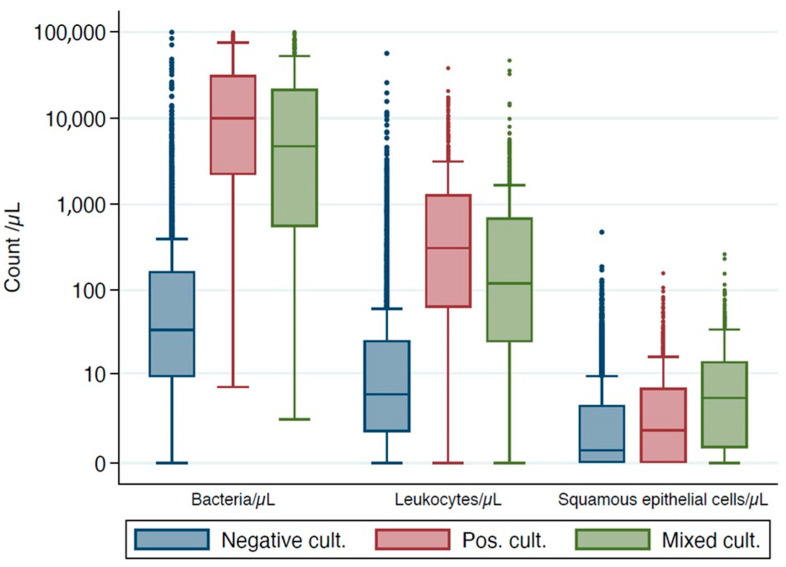
Boxplot diagrams of different UFC parameters according to the type of urine culture growth.

**Figure 3 diagnostics-12-01008-f003:**
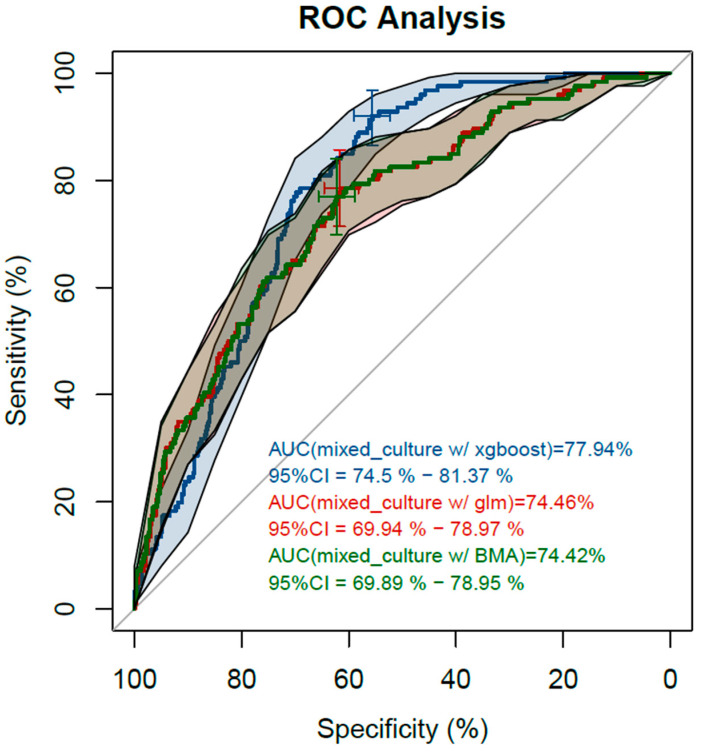
ROC with AUC and 95% CI. Blue curve: analysis using XGBoost with AUC 0.779 (95% CI: 0.745, 0.815). Red curve: analysis using glm with AUC 0.745 (95% CI: 0.699, 0.790). Green curve: analysis using BMA with AUC 0.744 (95% CI: 0.699, 0.790).

**Figure 4 diagnostics-12-01008-f004:**
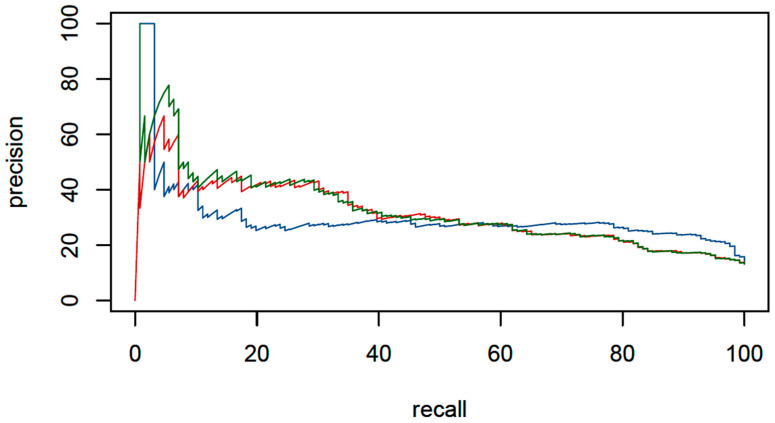
Precision–recall curves. The blue curve represents the analysis using the XGBoost package. For the red curve, glm was used, and for the green curve BMA was used. precision = PPV = (true positive)/(true positive + false positive). recall = sensitivity = (true positive)/(true positive + false negative).

**Figure 5 diagnostics-12-01008-f005:**
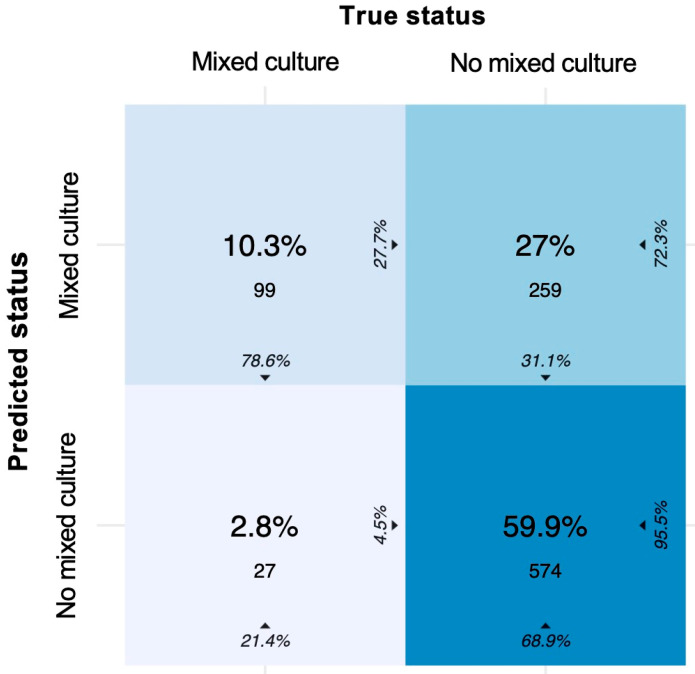
Confusion matrix of the two-class approach to predict mixed culture. Colour intensity represents the overall percentages in the middle of each tile. The column percentage is shown at the bottom, and the row percentage can be seen 90 degrees rotated to the right of each tile. In the diagonal from top left to bottom right, the column percentages represent the SE, and the row percentages represent the PPV of each class, e.g., in the upper left quadrant, 10.3% (*n* = 99) of all observations was predicted as mixed culture, where the true status was mixed culture. This corresponds to 78.8% of all true mixed cultures and 27.7% of all predicted mixed cultures.

**Figure 6 diagnostics-12-01008-f006:**
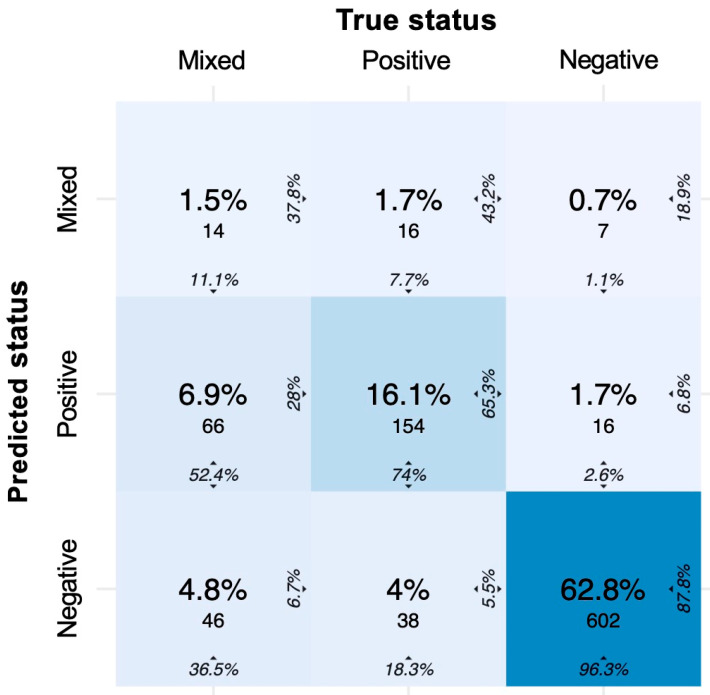
Confusion matrix of the tree-class approach to predict urine culture growth (negative, positive, mixed). Colour intensity represents the overall percentages in the middle of each tile. The column percentage is shown at the bottom, and the row percentage can be seen 90 degrees rotated to the right of each tile. In the diagonal from top left to bottom right, the column percentages represent the SE, and the row percentages represent the PPV of each class.

**Table 1 diagnostics-12-01008-t001:** Characteristics.

	Total (*n* = 3835)		Training Set (*n* = 2876)		Validation Set (*n* = 959)	
**Age [years], med (IQR)**	67	(51–78)	67	(50.5–78)	66	(51–78)
**Gender, *n* (%)**						
Male	2084	(54.3)	1548	(53.8)	536	(55.9)
Female	1751	(45.7)	1328	(46.2)	423	(44.1)
**Urine sample, *n* (%)**						
Single-use catheter urine	493	(12.9)	363	(12.6)	130	(13.6)
Midstream urine	2206	(57.5)	1639	(57.0)	567	(59.1)
Spontaneous urine	1136	(29.6)	874	(30.4)	262	(27.3)
**Bacteria in UFC/µL,** **med (IQR)**	150	(19–2814)	149	(18–2901)	150	(21–2711)
**Leucocytes in UFC/µL,** **med (IQR)**	17	(4–167)	17	(4–170)	18	(4–155)
**Squamous epithelial cells in UFC/µL, med (IQR)**	2	(0–6)	2	(0–6)	2	(0–7)
**Urine culture growth, *n* (%)**						
Negative culture						
Positive culture	2501	(65.2)	1876	(65.2)	625	(65.2)
*Escherichia coli*	552	(14.4)	420	(14.6)	132	(13.8)
*Klebsiella pneumoniae*	75	(2.0)	49	(1.7)	26	(2.7)
*Enterococcus faecalis*	18	(0.5)	12	(0.4)	6	(0.6)
*Aerococcus urinae*	16	(0.4)	12	(0.4)	4	(0.4)
*Staphylococcus aureus*	15	(0.4)	12	(0.4)	3	(0.3)
*Klebsiella oxytoca*	14	(0.4)	9	(0.3)	5	(0.5)
*Lactobacillus species*	14	(0.4)	9	(0.3)	5	(0.5)
*Pseudomonas aeruginosa*	11	(0.3)	5	(0.2)	6	(0.6)
Other	102	(2.7)	81	(2.8)	21	(2.2)
Mixed culture	517	(13.5)	391	(13.6)	126	(13.1)

**Table 2 diagnostics-12-01008-t002:** Univariable association of squamous epithelial cells and mixed culture in the validation set using logistic regression analysis.

Squamous Epithelial Cells Group/µL ^1^	Mixed Culture, *n* (%)		No Mixed Culture, *n* (%)		Odds Ratio (95% CI)	
0–0.1	2	(1.6)	64	(7.7)	1.00	(baseline)
>0.1–0.5	6	(4.8)	154	(18.5)	1.25	(0.25–6.34)
>0.5–1.9	27	(21.4)	254	(30.5)	3.40	(0.79–14.68)
>1.9–6.3	27	(21.4)	173	(20.8)	4.99	(1.15–21.61)
>6.3–17.5	32	(25.4)	113	(13.6)	9.06	(2.10–39.06)
>17.5	32	(25.4)	75	(9.0)	13.65	(3.15–59.20)

^1^ The group limits were obtained in the testing set by the 10th, 25th, 50th, 75th and 90th percentile of squamous epithelial cells.

**Table 3 diagnostics-12-01008-t003:** SE, SP, PPV and NPV for the prediction of mixed, positive and negative cultures.

**Prediction of Mixed Cultures**	**(%)**
Sensitivity (SE)	11.1
Specificity (SP)	97.2
Positive Predictive Value (PPV)	37.8
Negative Predictive Value (NPV)	87.9
**Prediction of Positive Cultures**	**(%)**
Sensitivity (SE)	74.0
Specificity (SP)	89.1
Positive Predictive Value (PPV)	65.3
Negative Predictive Value (NPV)	92.5
**Prediction of Negative Cultures**	**(%)**
Sensitivity (SE)	96.3
Specificity (SP)	74.9
Positive Predictive Value (PPV)	87.8
Negative Predictive Value (NPV)	91.6

## Data Availability

Data are available on request if the regulatory conditions for further use are met.

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
