# Peer review of "Urine Flow Cytometry Parameter Cannot Safely Predict Contamination of Urine—A Cohort Study of a Swiss Emergency Department Using Machine Learning Techniques"

_diagnostics, 2022, doi:10.3390/diagnostics12041008_

Round 1

Reviewer 1 Report

The most important weakness of the study is the absence of clinical data. Perhaps authors may perform a new study in the future combining clinical data with UFC findings

Author Response

We thank the reviewer for the review. We agree with the reviewer that the most important weakness of the study is the absence of additional clinical data. We stated this shortcomings in the limitation section in the fourth paragraph.

Reviewer 2 Report

The authors have conducted an interesting and well-structured study about urine flow cytometry analyses of urine samples in order to predict different types of urine culture growth, including mixed growth indicating urine culture contamination, and its possible practical clinical use.  

As known, the prevalence of bacteriuria, or bacteria in the urine, is more in adult women than in men. While this difference can be explained by anatomical differences between men and women, various evidence suggests that sex bias in UTIs can be also driven by sex-based variation in the levels of specific hormones, such as testosterone or estrogen, over the course of a lifetime. Additionally, it seems that the incidence of UTIs in geriatric populations is almost similar between men and women. Considering that the authors mentioned in their study that age and sex were evaluated as potential predictors, therefore, I suggest including in your discussion a small paragraph that includes these aspects in relation to your results.  

Row 374 correct “practise”

Author Response

We thank the reviewer for his/her careful review of our manuscript and the overall positive evaluation of our manuscript.

We followed the reviewer's suggestion and elaborated in the discussion section of the revised version of our manuscript about the association of the type of urine culture growth and sex respectively age. Furthermore, we added a sentence about the association of age and sex regarding the type of urine culture growth to the result section and created a supplement figure for further visualisation.